# Molecular Diagnosis and Vegetative Compatibility Group Analysis of Fusarium Wilt of Banana in Nepal

**DOI:** 10.3390/jof9020208

**Published:** 2023-02-05

**Authors:** Bimala Pant, Tingting Bai, Chanjuan Du, Suraj Baidya, Prem Bahadur Magar, Shrinkhala Manandhar, Jiban Shrestha, Miguel Dita, Mathieu Rouard, Gang Fu, Si-Jun Zheng

**Affiliations:** 1Nepal Agricultural Research Council, National Plant Pathology Research Centre, Lalitpur 44700, Bagmati Province, Nepal; 2Yunnan Key Laboratory of Green Prevention and Control of Agricultural Transboundary Pests, Agricultural Environment and Resources Institute, Yunnan Academy of Agricultural Sciences, Beijing Road 2238, Kunming 650205, China; 3Guangxi Key Laboratory of Biology for Crop Diseases and Insect Pests, Plant Protection Research Institute, Guangxi Academy of Agricultural Sciences, Daxue Road 174, Nanning 530007, China; 4Department of Entomology and Plant Pathology, North Carolina State University, Raleigh, NC 27695, USA; 5Nepal Agricultural Research Council, National Plant Breeding and Genetics Research Centre, Lalitpur 44700, Bagmati Province, Nepal; 6Bioversity International, Cali-Palmira CP, Palmira 763537, Colombia; 7Bioversity International, Parc Scientifique Agropolis II, CEDEX 5, 34397 Montpellier, France; 8Bioversity International, Beijing Road 2238, Kunming 650205, China

**Keywords:** Fusarium wilt of banana, *Foc* race 1, pathogen identification, Panama diseases, vegetative compatibility group

## Abstract

Fusarium wilt of banana (FWB), caused by *Fusarium oxysporum* f. sp. *cubense* (*Foc*), is the most important constraint of the banana industry globally. In Nepal, epidemics resembling FWB have been increasingly observed on the Malbhog cultivar in the past several years. However, the disease has not been officially reported yet, and consequently, little is known about the pathogen present across the country. In this study, we characterized 13 fungal strains isolated from banana plants of the Malbhog cultivar (Silk, AAB) showing symptoms similar to FWB in banana plantations in Nepal. All of the strains were typed as belonging to the *F. oxysporum* and caused FWB symptoms when inoculated in the Malbhog and Cachaco (Bluggoe, ABB) cultivars. No symptoms were observed in the Williams cultivar (Cavendish, AAA). Vegetative compatibility group (VCG) analysis classified the strains as VCG 0124 or VCG 0125. PCR analyses conducted with primers specific for *Foc* race 1 (*Foc* R1) or *Foc* tropical race 4 (TR4) revealed that all the strains reacted positively for *Foc* R1 and none for TR4. Altogether, our results demonstrated that the pathogen populations causing FWB of the Malbhog cultivar in Nepal were *Foc* R1. This work reported, for the first time, the occurrence of FWB in Nepal. Further studies with larger *Foc* populations are needed to better understand disease epidemiology to design sustainable disease management strategies.

## 1. Introduction

Banana is a major fruit crop and a high-value agricultural product in Nepal. It is cultivated on 18,329 ha areas across the country, with a total production of 278,890 tons and a productivity of 16.79 t/ha spanning 68 out of the 77 districts of Nepal [1,2]. Among them, Chitwan district is considered the major hub for banana production, followed by the Saptari, Jhapa, Morang, and Rupandehi districts. In Chitwan, banana farming is mainly concentrated in Bharatpur, Ratnanagar, Kalika, Khaireni, Jagatpur, and Thimura [3]. Nearly 15% of the area of banana-growing regions in Nepal is covered by a variety called Malbhog (Silk, AAB), which accounts for a 2493 ha area with 41,833 tons of production [4].

Diseases are major constraints to the banana production in Nepal. Fusarium wilt of banana (FWB), also known as Panama disease, is caused by the soil-borne fungus *Fusarium oxysporum* f. sp. *cubense* (*Foc*), which is a growing concern in Nepal. *Foc* was first described in Australia in 1874 [5]. Due to the exchange of planting material and the movement of spore-bearing soil, the pathogen has spread and has been reported in many countries of the world [6,7]. So far, three races of *Foc* (R1, R2, and R4) have been reported, based on the pathogenic characterization of different banana cultivars [8]. *Foc* R1, which caused FWB epidemic in the early 20th century, affects a range of cultivars, such as the Gros Michel (AAA), Silk (AAB), and Pisang Awak (ABB) subgroups, among others. *Foc* R2 affects cooking bananas belonging to the Bluggoe (ABB) subgroup. *Foc* R4, which is further divided into subtropical race 4 (SR4) and tropical race 4 (TR4), affects Cavendish cultivars, as well as bananas susceptible to *Foc* R1 and *Foc* R2.

Apart from races, *Foc* has been also grouped based on vegetative compatibility (VCGs). So far, 24 VCGs have been identified, but there is solid evidence that other VCGs also exists [9]. By far, TR4, which belongs to VCG 01213/16, is the most aggressive strain causing FWB. The strain, originating from Indonesia, has spread globally on different continents [10,11,12,13,14]. The strain is currently under strict quarantine regulations in most of the banana-producing countries worldwide. Therefore, the early detection of *Foc* TR4 is overarching, as it represents a serious threat for the export banana industry, banana diversity, and food security. *Foc* R1 strains belonging to VCG 0124 and VCG 0125 were also reported to infect the Cavendish banana variety, Grand Naine banana, under specific circumstances in India [15,16]. SR4 comprises strains belonging to VCG 0120, 0121, 0122, 0129, and VCG 01211 [17]. All the other VCGs, including VCG 0124 and VCG 0125, belong to *Foc* R1 or *Foc* R2 [17,18,19]. The VCG classification is imperfect; it is noted that VCG 0124 belongs to *Foc* R1, as well as *Foc* R2, while the VCG 0120 strain can be classified as both *Foc* R1 and SR4 [10,18,20]. This is considered to be the result of convergent evolution or the horizontal gene transfer of the *F. oxysporum* [10,21,22].

A syndrome similar to FWB appeared in banana plantations of Nepal in 2017 in the Chitwan district. It became widespread and destructive in almost all Malbhog banana-growing regions of Nepal, with a 30–90% disease incidence. In Chitwan, some farmers had to abandon the Malbhog variety, due to the high susceptibility to this disease. While field observations indicated that the disease corresponded to FWB, it was not clear which strain was associated with it. Both farmers and phytosanitary authorities were concerned about an eventual outbreak caused by *Foc* TR4 in Nepal because TR4 was already reported both in Bihar and Uttar Pradesh, India, which borders Nepal [23,24]. In the present study, we isolated the causal agent responsible for the disease outbreak in Malbhog bananas in Nepal, investigated the vegetative compatibility, and tested the pathogenicity to facilitate the development of effective disease management strategies in Nepal.

## 2. Materials and Methods

### 2.1. Sample Collection

In December 2017, an unusual phenomenon of banana plants (Malbhog, AAB) showing severe yellowing wilt symptoms before collapse was first reported in the Chitwan district of Nepal. Plant sampling from Malbhog bananas was conducted during two field surveys in December 2019 in the Chitwan and Nawalparasi districts. Visited locations were at an altitude of 170 m to 210 m. Diseased plants with typical external symptoms were found, and some xylem vessels appeared reddish-brown when the pseudostems were cut longitudinally. Altogether, thirteen samples from symptomatic tissue were collected from seven farmers’ fields of the Chitwan district and four farmers’ fields of the Nawalparasi district (Table 1).

### 2.2. Isolation and Identification

All collected samples were dried at room temperature before the isolation of the pathogen. Every sample was cut into small squares with a width of 0.5 cm using scissors. The scissors and tweezers were repeatedly heat-sterilized for every sample isolation. To prevent cross-contamination, gloves were changed when handling each sample. The cut pieces were immersed in 70% alcohol for 30 s for disinfection, followed by immersion in 0.1% mercury chloride for 30 s in the same way, and then rinsed three times with sterile water and placed on the plates with PDA medium with 50 µg/mL ampicillin. Then, the plates were kept in the dark at 30 °C for 5–7 days. The morphologies of the developed colonies were observed under a stereomicroscope, and the typical structures of *Fusarium* were observed using a compound microscope.

### 2.3. Pathogenicity Test

Tissue-cultured plantlets of Malbhog were planted in a steam-sterilized potting medium consisting of soil, sand, and farmyard manure in the ratio of 3:1:1 at the National Plant Pathology Research Centre (NPPRC), Khumaltar, Nepal. The pathogenicity test was performed when the plants reached 10–15 cm in height at the 3–4 leaf stage to ensure that the vascular system was fully developed. Six plants were used per treatment. These plants were inoculated with NP11 or NP12 single-spore isolates from the Chitwan and Nawalparasi districts, respectively. The conidial suspension adjusted was counted using a hemocytometer and was 6.3 × 10^6^ conidia/mL. The roots of plants were slightly wounded by a sterile knife, and 100 mL of spore suspension of the *F. oxysporum* isolate was then added to the plantlets by drenching. Plants drenched with water served as the control. All the plants were placed in the greenhouse. Inoculated plants were monitored regularly for the appearance of disease symptoms. When external symptoms appeared, the internal symptoms were also investigated by dissecting the pseudostem and corm. The pathogen was re-isolated from the inoculated plants and further characterized by PCR.

Pathogenicity tests were also carried out at the Institute of Plant Protection Research, Guangxi Key Laboratory of Biology for Crop Diseases and Insect Pests, Guangxi Academy of Agricultural Sciences, China. Banana plantlets from the Cachaco (Bluggoe, ABB) and Williams (Cavendish, AAA) cultivars were grown and inoculated as afore mentioned. Nine plants were used per treatment, and each treatment was repeated three times. The wounded roots of the inoculated plantlets were soaked in a 10^6^ conidia/mL spore suspension for 30 min and then planted in the pots. Those plantlets that soaked in sterile water for 30 min served as the healthy control. Twenty-four days after inoculation, the disease index was investigated by dissecting the corm. The classification criteria were as follows: scale 0, no symptoms in the corm; scale 1, discoloration area in the central longitudinal section of the corm was less than 25%; scale 3, discoloration area in the central longitudinal section of the corm was 26–50%; scale 5, discoloration area in the central longitudinal section of the corm was 51–75% area; scale 7, discoloration area in the central longitudinal section of the corm was more than 75%. The disease index was calculated according to the following equation [25]:Disease index equation = [Σ (number of infected plants at a given disease scale × the disease scale)/(total tested plants × 7)] × 100

Statistical analysis was performed using R Studio version 4.0.3 (Agricolae package) for the disease index data. The data was transformed into log^10^ (1 + data) for normalization, and 1 in each data set was added to remove potential infinity values from the log-transformed data. The analysis was measured using two-way ANOVA with the LSD test.

### 2.4. Vegetative Compatibility Group Analyses

Nitrate-nonutilizing (nit) mutants of the wild-type *Foc* strains were generated in minimal medium (MM) [26] amended with KClO_3_ and incubated for 7–14 days at 25 °C. Spontaneous KClO_3_-resistant sectors were transferred to MM. Those that grew as thin colonies with no aerial mycelium were classified as nit mutants and were further characterized on media containing one of four different sources of nitrogen [27]. Finally, VCGs of all mutants were determined by pairing in MM with tester nit mutants from strains with known VCGs (VCG 0120, VCG 0121, VCG 0122, VCG 0123, VCG 0124, VCG 0125, VCG 0126, VCG 0127, VCG 0128, VCG 0129, VCG 01210, VCG 01211, VCG 01212, VCG 01213, VCG 01214, VCG 01215, VCG 01216, VCG 01217, VCG 01218, VCG 01219, VCG 01220, VCG 01221, VCG 01222, and VCG 01223) obtained from Professor Elizabeth Aitken, the University of Queensland, Australia. Complementation between different nit mutants resulted in dense aerial growth at the contact zone between the two colonies [27]. Eight isolates (NP1-1, NP2-1, NP4-1, NP5-1, NP6-1, NP7-1, NP8-1, and NP10-1) were used for vegetative compatibility group analyses.

### 2.5. PCR Analyses

Five- to seven-day-old cultures were used for polymerase chain reaction (PCR) amplification. Approximately 100 mg of mycelium was taken with a sterile toothpick and placed into a sterilized PCR tube with 50 µL of lysis buffer for microorganisms (Takara Bio, https://www.takarabio.com, accessed on 13 January 2020). After 15 min of thermal denaturation at 80 °C in the PCR machine, it was centrifuged briefly, and 1.5 µL of the supernatant was used as a template for the PCR reaction. The translation elongation factor 1α gene (TEF-1α) was amplified with primers EF-1 and EF-2 as internal positive controls using the following program: 95 °C for 2 min and 35 cycles of 95 °C for 30 s; 50 °C for 30 s; and 72 °C for 1 min, followed by an additional extension time for 10 min at 72 °C [21]. The primer sets W1805F/W1805R and W2987F/W2987R from Li et al. (2012) [20] were used to identify *Foc* R1 (354 bp) and *Foc* R4 (593 bp), respectively. Another pair of W2987F/W2987R from Li et al. (2013) [28] was used to identify TR4 (452 bp). [Note: the name of primer set W2987F/W2987R from Li et al. (2013) [28] is the same as that from Li et al. (2012) [20], so in this research, we labeled it as TR4-W2987F/TR4-W2987R.] The sequence information of all the primer sets is shown in Table 2. The PCR was performed using the Gold Mix (Code: TSE 101, tsingke, Beijing, China) according to the manufacturer’s protocol.

PCR products were electrophoresed in 1% agarose gel in an electrophoretic tank (Takara Bio) containing 1X TAE buffer for 20 min at 135 V. BIO-RAD ChemiDocTMXRS+ was used to take an image after electrophoresis.

### 2.6. Phylogenetic Analysis

The TEF-1α sequences of isolates were amplified by PCR. These PCR products were confirmed with aimed bands by agarose gel electrophoresis and purified using the DNA Gel Extraction kit TSP601 (Tsingke, Beijing, China). PCR products were then cloned and sequenced in the pMD18-T vector. The TEF-1α sequences were showed in Appendix A. Additionally, TEF-1α sequences of other *F. oxysporum* isolates were obtained from the publicly available genomes (https://www.ncbi.nlm.nih.gov/, accessed on 30 July 2021). The analysis involved 31 isolate sequences that were aligned with MUSCLE [29], and the phylogenetic analyses were performed using maximum likelihood (ML). The phylogenetic tree was constructed using the MEGA 6.0 software with the Kimura 2-parameter model [30]. The TEF-1α sequence from *Ramlispora sorghi* was used as the outgroup taxon (KM087651.1). Branches were tested for the inferred tree by bootstrap analysis on 1000 random trees.

## 3. Results

### 3.1. Disease Diagnosis with Field Symptoms

The problem of wilt disease has existed in banana orchards in Nepal since 2017. The disease was observed in some new and old orchards growing the Malbhog variety. Symptoms appeared as yellowing of the oldest leaves and splitting at the base of different types of infected plants. Some diseased, old yellow leaves split at the base. Vascular discoloration varied from pale yellow in the early stage to dark red or almost black in the later stage when viewed by cutting a cross-section or splitting the pseudostem base. Brownish streaks of the vascular tissue were also observed during the field survey (Figure 1). However, wilt disease was not observed in the Grand Naine banana variety.

### 3.2. Morphology of the Pathogen Isolate

Thirteen suspected pathogen isolates were extracted from thirteen corresponding samples collected from banana fields in the Chitwan and Nawalparasi districts (Table 1), and their morphological characteristics were studied. The isolates produced dense white and purple colonies, and mycelia were evenly spread all over the PDA medium (Figure 2A,B).

The microscopic examination of fungal isolates showed the presence of three types of asexual spores: macroconidia, microconidia, and chlamydospores. The macroconidia were thin-walled with a definite foot cell and a pointed apical cell (Figure 2C). Microconidia were oval- or kidney-shaped and were produced on micro-conidiophores in aerial mycelia (Figure 2D). Chlamydospores were thick-walled (Figure 2E), and these were usually produced singly in macroconidia, were intercalary, or were in terminal hyphae [31].

### 3.3. Pathogenicity Test

The pathogenicity test was performed on three banana cultivars: Malbhog (AAB), Cachaco (ABB), and Williams (AAA). Yellowing and wilting symptoms were observed in inoculated Malbhog plantlets after 21 days of pathogen inoculation (Figure 3A: left two plants). The non-inoculated plants (controls) showed no symptoms, and the leaves remained green (Figure 3A: right two plants). Dark spots and brownish discoloration of the vascular tissue in the pseudostem were also observed (Figure 3B). In severe cases, inoculated plantlets showed complete wilting or even death.

Similar symptoms were observed in inoculated Cachaco plantlets (Figure 4A,C). Wilting symptoms in plantlets inoculated with the isolates NP1-1, NP2-1, NP4-1, NP5-1, NP6-1, NP7-1, NP8-1, and NP10-1 were noticed after 24 days of pathogen inoculation. There was a highly significant response of the disease index on the varieties and interactions of isolates with varieties, while the response was non-significant in the case of isolates without varieties (Table 3). The pathogenicity of the *F. oxysporum* isolates to Cachaco and Williams was extremely significantly different (*p* < 2 × 10^−16^). All of the inoculated Cachaco plantlets showed different degrees of symptoms, and the disease index was greater than 26.19, while the inoculated Williams plantlets nearly had no disease symptoms. The disease index of Cachaco inoculated with the isolate NP4-1 was 66.67, which was significantly higher than that caused by NP2-1, NP5-1, NP6-1, and NP8-1 (*p* < 0.0465). The pathogen was re-isolated from the inoculated, symptomatic plants and confirmed with the original one using microscopic observation and PCR.

### 3.4. PCR Analysis

PCR analyses were performed using four primer sets (Figure 5). The W1805F/W1805R primer set previously associated with *Foc* R1 populations [20] produced the predicted 354 bp amplicon in all of the 13 tested isolates and from the *Foc* R1 strain used as the reference control. None of the tested isolates reacted to the *Foc* R4 or TR4 primer sets. The TR4 strain used as the positive control produced the predicted 593 and 452 bp amplicons, respectively. Altogether, our PCR results suggested that all *Foc* strains isolated from Malbhog bananas in Nepal were associated with *Foc* R1 populations.

### 3.5. Vegetative Compatibility Analyses

Nit mutants were successfully generated for eight *Foc* isolates (NP1-1, NP2-1, NP4-1, NP5-1, NP6-1, NP7-1, NP8-1, and NP10-1). Five (NP2-1, NP4-1, NP7-1, NP8-1, and NP10-1) out of the eight isolates were compatible with VCG 0124, since the development of aerial mycelia and the formation of heterokaryons at the line of intersection between colonies of the isolates and tester mutants were observed. The other isolate (NP5-1) was typed as VCG 0125, suggesting that VCG 0124 was the predominant isolate. NP1-1 and NP6-1 were of unknown VCGs (Figure 6 and Figure 7).

### 3.6. Phylogenetic Analysis

Nepal isolates and some other *F. oxysporum* isolates from the publicly available genomes were phylogenetically analyzed based on TEF-1α. The phylogenetic tree divided thirteen Nepal isolates into two clusters. The first cluster, including most of the isolates, viz., NP1-1, NP2-1, NP5-1, NP6-1, NP7-1, NP8-1, NP11, NP12, NP13, NP14, and NP15 were grouped with *Foc* NRRL 25607 and CAV 2612, which were also typed as VCG 0124 in previous studies [18,32]. The second cluster grouped NP4-1 and NP10-1 together with a set of *F. oxysporum* formae speciales, including *cubense* (NRRL 25367 NRRL 25609) and non-*cubense,* such as *glycines* (NRRL 25598) and *melonis* (NRRL 26178) (Figure 7).

## 4. Discussion

FWB causes severe yield losses globally, especially the TR4 strain, which is considered the biggest threat to global banana production [33]. Since 2017, typical symptoms of FWB have been increasingly observed in banana plantations in Nepal. However, identification and characterization of the causal agent is still lacking. In this research, we isolated and characterized the pathogen responsible for causing wilt disease on Malbhog banana plantations in the Chitwan and Nawalparasi districts, Nepal. Isolates, which exhibited morphological characteristics similar to the *Fusarium oxysporum* [34], were able to cause typical FWB symptoms in the Malbhog and Cachaco cultivars but not in the Cavendish cultivar, Williams. Field observations also revealed that the Cavendish cultivar, Grand Naine, which is resistant to *Foc* R1, had successfully replaced the susceptible Malbhog in the affected areas. Our PCR and VCG analyses grouped all the strains isolated as *Foc* R1 and revealed that six of these strains belonged to VCGs 0124 and 0125.

The phylogenetic tree based on TEF-1α divided the thirteen Nepal isolates into two clusters. The first cluster included most isolates grouped with *Foc* NRRL 25607 and CAV 2612, both of which belonged to VCG 0124 [18,32]. NP5-1, which belonged to VCG 0125, was also grouped with them. NP4-1 and NP10-1, both of which belonged to VCG 0124, were grouped into a separate branch. Results of genetic diversity derived from VCG and the phylogenetic tree were not fully consistent. It was probably due to the complex genetic diversity of *Foc* for convergent evolution or the horizontal gene transfer of *F. oxysporum* [10,21,22]. Moreover, TEF-1α was shown in some cases to be insufficient for distinguishing different formae speciales of *Fusarium* [35]. Nonetheless, results of the VCG revealed the diversity of *Foc* R1 in Nepal, although a larger number of isolates collected from other banana-producing regions may be suitable to further understand the *Foc* population biology in Nepal.

Banana is a high-value agricultural product and a major fruit in Nepal, in terms of production and domestic consumption (https://www.icimod.org/expanding-commercial-banana-production-in-nepal/, accessed on 8 June 2021). The current epidemic and spread of *Foc* R1 have severely affected the production of the traditional banana cultivar, Malbhog, which is one of the preferred local cultivars in Nepal. Our surveys revealed that FWB was prevalent in almost 90% of the Malbhog-growing area of the Chitwan and Nawalparasi districts. Surveys and farmer interviews during field visits indicated that such a rapid spread of FWB could be associated with the informal exchange of planting material from infested to healthy areas. In the absence of effective management practices, many growers chose to plant the *Foc* R1-resistant cultivar, Grand Naine. However, farmers and extension officers need to consider that Grand Naine is highly susceptible to *Foc* TR4, a more aggressive *Foc* strain already present in Bihar and Uttar Pradesh, India [23,24]. Therefore, prevention and exclusion measures against *Foc* TR4 using anticipatory approaches, as well as continuous *Foc* population monitoring, are strongly encouraged.

## 5. Conclusions

In conclusion, 13 fungal strains were isolated from banana plants of the Malbhog cultivar (Silk, AAB) showing symptoms similar to FWB in banana orchards in Nepal. Molecular diagnosis and vegetative compatibility group analysis were conducted for the evaluation of the FWB. All the strains caused FWB symptoms when inoculated in the Malbhog and Cachaco (Bluggoe, ABB) cultivars. No symptoms were observed in the Williams cultivar (Cavendish, AAA). PCR analyses with primers specific for *Foc* race 1 (R1) or *Foc* tropical race 4 (TR4) revealed that all the strains reacted positively for *Foc* R1 and none for TR4 in this study. Vegetative compatibility group (VCG) analysis further confirmed most strains as VCG 0124 or VCG 0125. The results demonstrated that the pathogen in this study causing FWB in Nepal are associated with *Foc* R1 in these limited samples. Larger *Foc* populations are necessary to better understand disease epidemiology for making sustainable disease management strategies in the next step.

## Figures and Tables

**Figure 1 jof-09-00208-f001:**
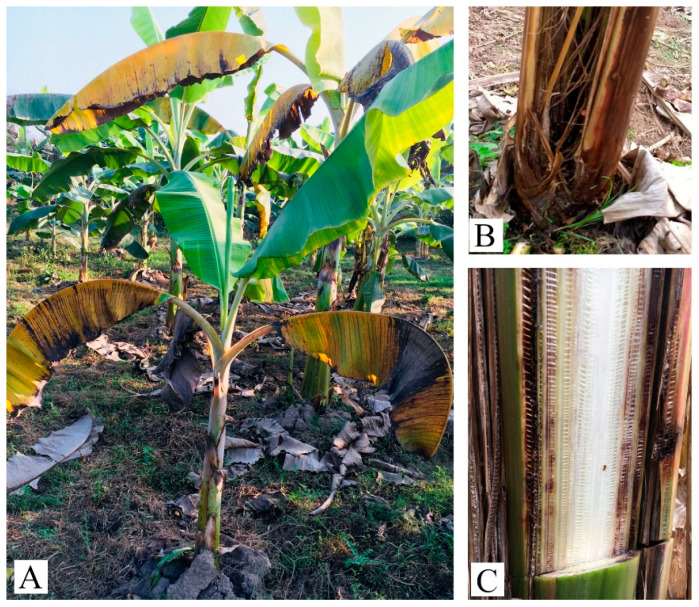
Different diseased plants (Malbhog variety) with various symptoms on different parts in the Chitwan district. (**A**) Whole plant, (**B**) splitting pseudostem, and (**C**) reddish to dark-brownish discoloration of the vascular system.

**Figure 2 jof-09-00208-f002:**
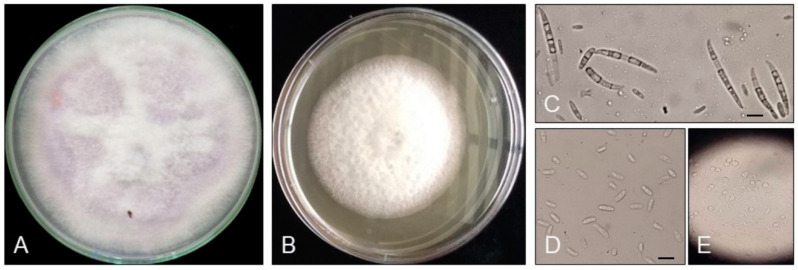
Morphological characteristics of the fungus isolated from banana orchards of the Chitwan and Nawalparasi districts. (**A**) Purple shades of *Fusarium oxysporum* f. sp. *cubense* (*Foc*) on PDA medium; (**B**) white shades of *Foc* on PDA medium; (**C**) macroconidia of *Foc*; (**D**) microconidia of *Foc*; and (**E**) chlamydospores of *Foc*. All scale bars 10 μm.

**Figure 3 jof-09-00208-f003:**
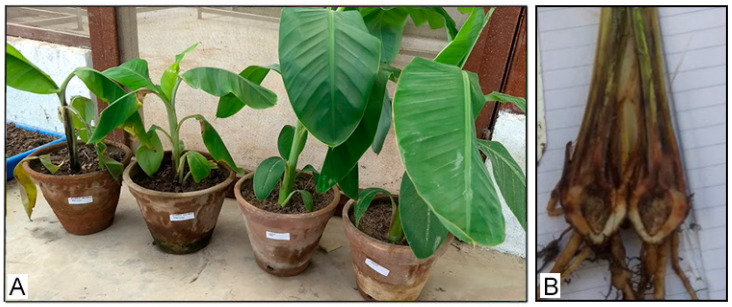
Pathogenicity test on the Malbhog (AAB) variety at the National Plant Pathology Research Center, NARC, Khumaltar, Nepal. (**A**) Inoculated (left two plants) and non-inoculated as control (right two plants); (**B**) rhizome discoloration caused by *Foc* isolates after inoculation.

**Figure 4 jof-09-00208-f004:**
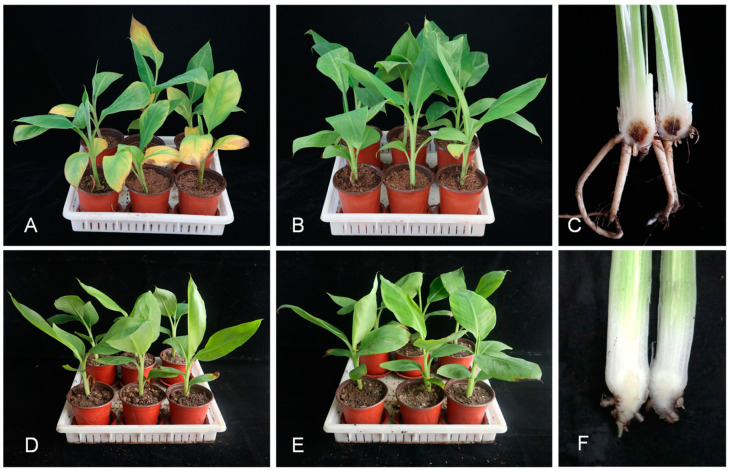
Pathogenicity test results on the Cachaco (ABB) and Williams (AAA) cultivars at the Institute of Plant Protection Research, Guangxi Key Laboratory of Biology for Crop Diseases and Insect Pests, Guangxi Academy of Agricultural Sciences, China. (**A**–**C**) Tests on Cachaco (ABB) cultivars; (**A**) inoculated; (**B**) non-inoculated; and (**C**) rhizome discoloration caused by *Foc* isolates after inoculation. (**D**–**F**) Tests on Williams (AAA) cultivars; (**D**) inoculated; (**E**) non-inoculated; and (**F**) no symptoms in the rhizome after inoculation.

**Figure 5 jof-09-00208-f005:**
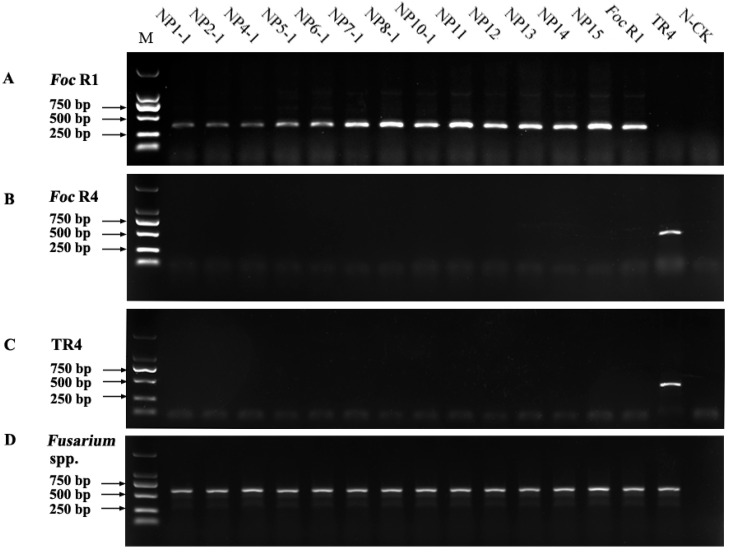
PCR identification of *F. oxysporum* f. sp. *cubense* (*Foc*) using primer sets (Table 2). (**A**) Primer set W1805F/W1805R; (**B**) Primer set W2987F/W2987R; (**C**) Primer set TR4-W2987F/TR4-W2987R; (**D**) Primer set EF-1/EF-2 was used to amplify the fragment of the translation elongation factor gene (TEF-1α) as the internal positive control. Lane M: molecular weight maker; NP1-1, NP2-1, NP4-1, NP5-1, NP6-1, NP7-1, NP8-1, NP10-1, NP11, NP12, NP13, NP14, and NP15 were the isolates from Nepal. The *Foc* R1 and TR4 isolates were from the Yunnan province of China; N-CK: water as negative control.

**Figure 6 jof-09-00208-f006:**
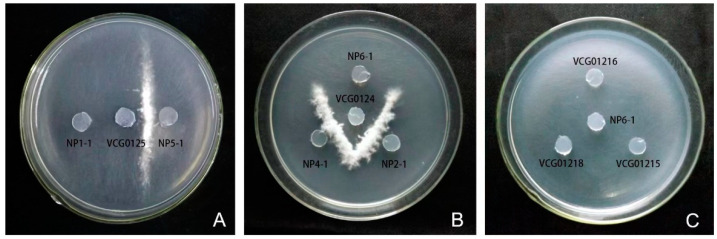
Compatibility assays between nit mutants on minimal medium (MM). (**A**,**B**) Heterokaryon formation when pairing the tester and some isolates; (**C**) no heterokaryon formation when pairing testers and isolate NP6-1.

**Figure 7 jof-09-00208-f007:**
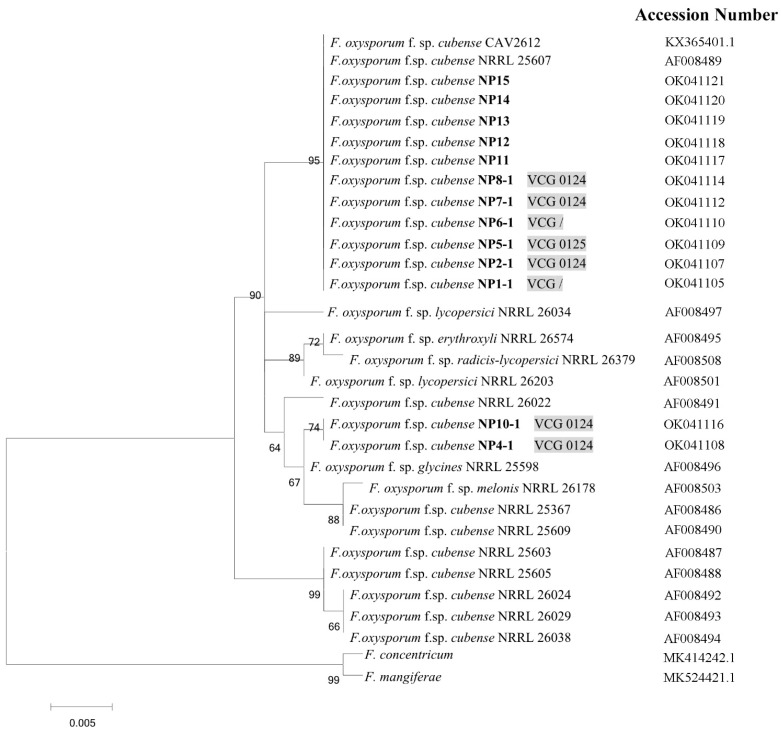
Maximum-likelihood (ML) tree inferred from the translation elongation factor 1α gene (TEF-1α) sequences in selected *Fusarium* isolates. Isolates obtained in the present study are indicated in bold. Bootstrap values (≥50%) are indicated at the nodes. The phylogram is rooted to *F. concentricum* M8164R and *F. mangiferae* SJG3-4. NRRL 25607 belongs to VCG 0124, associated with *Foc* R2 [18]. CAV 2612 also belongs to VCG 0124, but it is associated with *Foc* R1 [32]. VCG / means unidentified VCG.

**Table 1 jof-09-00208-t001:** Isolates of *Fusarium oxysporum* used in this study from the Chitwan and Nawalparasi districts of Nepal.

Isolate	Sampling Date	Site	Variety	Location	Altitude (m)
NP1-1	18 December 2019	Chitwan	Malbhog	27°40′55″ N, 84°30′21″ E	210
NP2-1	18 December 2019	Chitwan	Malbhog	27°40′55″ N, 84°30′24″ E	210
NP4-1	18 December 2019	Chitwan	Malbhog	27°40′55″ N, 84°30′24″ E	210
NP5-1	18 December 2019	Chitwan	Malbhog	27°40′55″ N, 84°30′24″ E	210
NP6-1	18 December 2019	Chitwan	Malbhog	27°40′55″ N, 84°30′24″ E	210
NP7-1	18 December 2019	Chitwan	Malbhog	27°37′50″ N, 84°31′1″ E	200
NP8-1	18 December 2019	Nawalparasi	Malbhog	27°37′3″ N, 84°6′1″ E	170
NP10-1	18 December 2019	Nawalparasi	Malbhog	27°37′3″ N, 84°6′1″ E	170
NP11	10 December 2019	Chitwan	Malbhog	27°40′52″ N, 84°30′26″ E	220
NP12	10 December 2019	Nawalparasi	Malbhog	27°37′3″ N, 84°6′1″ E	170
NP13	10 December 2019	Chitwan	Malbhog	27°39′12″ N, 84°30′41″ E	210
NP14	10 December 2019	Chitwan	Malbhog	27°39′10″ N, 84°30′41″ E	210
NP15	10 December 2019	Chitwan	Malbhog	27°37′52″ N, 84°31′4″ E	200

**Table 2 jof-09-00208-t002:** The primer sets used to identify the *Foc* pathogen.

Primer Sets	Fungi to Identify	Primer Sequences	References
W1805F/W1805R	*Foc* R1	5′-GTTGAGTCTCGATAAACAGCAAT-3′	[20]
		5′-GACGAGGGGAGATATGGTC-3′	
W2987F/W2987R	*Foc* R4	5′-GCCGATGTCTTCGTCAGGTA-3′	[20]
		5′-CTGAGACTCGTGCTGCATGA-3′	
W2987F/W2987R (TR4-W2987F/TR4-W2987R)	TR4	5′-TGCCGAGAACCACTGACAA-3′	[28]
		5′-GCCGATGTCTTCGTCAGGTA-3′	
EF-1/EF-2	*Fusarium* spp.	5′-ATGGGTAAGGA(A/G)GACAAGAC-3′	[21]
		5′-GGA(G/A)GTACCAGT(G/C)ATCATGTT-3′	

**Table 3 jof-09-00208-t003:** The disease index of isolates on the host.

Isolate	Variety	Disease Index (%)
	Cachaco (ABB)	38.9 (1.59a)
	Williams (AAA)	0.53 (0.11b)
*p*-value		<2 × 10^−16^
NP1-1		23.81 (0.84)
NP2-1		17.06 (0.85)
NP4-1		33.33 (0.91)
NP5-1		14.29 (0.73)
NP6-1		13.10 (0.70)
NP7-1		26.19 (0.98)
NP8-1		17.06 (0.97)
NP10-1		20.24 (0.81)
*p*-value		0.11
NP1-1	Cachaco (ABB)	47.62 (1.68abc)
NP2-1	Cachaco (ABB)	33.33 (1.52bc)
NP4-1	Cachaco (ABB)	66.66 (1.83a)
NP5-1	Cachaco (ABB)	28.57 (1.47bc)
NP6-1	Cachaco (ABB)	26.19 (1.41c)
NP7-1	Cachaco (ABB)	50.79 (1.71ab)
NP8-1	Cachaco (ABB)	31.75 (1.51bc)
NP10-1	Cachaco (ABB)	40.48 (1.61abc)
NP1-1	Williams (AAA)	0.00 (0.00e)
NP2-1	Williams (AAA)	0.79 (0.18de)
NP4-1	Williams (AAA)	0.00 (0.00e)
NP5-1	Williams (AAA)	0.00 (0.00e)
NP6-1	Williams (AAA)	0.00 (0.00e)
NP7-1	Williams (AAA)	1.59 (0.25de)
NP8-1	Williams (AAA)	2.38 (0.43e)
NP10-1	Williams (AAA)	0.00 (0.00e)
*p*-value		0.0465

Values are the means of three replications. Data were transformed using log^10^ (1 + data), the presented means are original, and the values inside the parentheses indicate transformed data; ANOVA was conducted on the transformed data. The mean values in the same column with the same letter are not statistically significantly different (*p* < 0.05).

## Data Availability

See Appendix A.

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
