# Peer review of "Molecular Diagnosis and Vegetative Compatibility Group Analysis of Fusarium Wilt of Banana in Nepal"

_jof, 2023, doi:10.3390/jof9020208_

Round 1
Reviewer 1 Report (Previous Reviewer 1)
The manuscript reports important first detection of Fusarium oxysporum f.sp. cubense in Nepal along with its characterization of races and VCG, that could help farmers in management of FWB in this country. The revised version following reviewers’ suggestions improved the manuscript that is now ready for publication after very minor corrections reported in the attached revised version of the manuscript.

Author Response
Please see the attachment.

Reviewer 2 Report (Previous Reviewer 2)
The authors exaustively answered to the comments and the manuscript has been improved so, in my opinion, it is now suitable for publication on JoF.
Author Response
Please see the attachment.

Reviewer 3 Report (New Reviewer)
The revised manuscript solves reads better and answered reviewer's questions.
Author Response
Please see the attachment.

This manuscript is a resubmission of an earlier submission. The following is a list of the peer review reports and author responses from that submission.
Round 1
Reviewer 1 Report
The manuscript reports important first detection of Fusarium oxysporum f.sp. cubense in Nepal that could help farmers in management of FWB in this country. However, results obtained would perhaps fit better in a disease note or in a completely revised version of the manuscript.
Results are preliminary as they state themselves: isolates are few, molecular analyses are incomplete: more genes must be analyzed for taxonomy and primers used are not clear, pathogenicity tests must be better explained (see Fig.4).
Also the text have to be profoundly revised: there are wrong captions, wrong paragraphs, double references and so on (showed in the text).
To see details please refer to corrections and comments reported in the revised version of the manuscript.

Reviewer 2 Report
The paper of Pant and colleagues entitled “Molecular diagnosis and vegetative compatibility group analysis of Fusarium wilt of banana in Nepal”, reports for the first time the occurrence of FWB in Nepal. The authors isolated from Malbhog cultivar 13 strains, which were molecularly identified as belonging to the F. oxysporum species complex, Foc race 1 (R1). They conclude that the pathogen populations causing FWB in Nepal are associated to R1 and discard the presence of Foc tropical race 4 (TR4), the most aggressive strain causing FWB. In my opinion, 13 strains isolated from one cultivar represent just one population of the pathogen and are absolutely inadequate to represent the Foc populations present in Nepal and to conclude that the Foc race 1 is the only one present in banana growing areas. The authors state that “further studies with larger Foc populations are needed to better understand disease epidemiology to design sustainable disease management strategies”. I think this need is, first of all, important to have a clear picture of the races of this destructive pathogen present in Nepal.
Moreover, the paper is not fluent and easily readable: figures, figure legends and the text are not congruent in several cases, some important details are lacking in Material and Methods and are confused, results are confused. Names of fungal species should be in Italics.
I believe the paper is not suitable for publication on JoF in this version. Here some suggestions for the authors:
Line 39: replace “confirms” with “report”;
Line 54-56: check the punctuation, the sentence is unclear;
Line 63: delete which;
Line 109: maybe 0,1% HgCl2 ;
Line 119: two plants are not sufficient for testing pathogenicity;
Line 122 and 134: replace 106 with 106;
Line 124-126: where do you put plants after drenching?;
Line 136-140: this should be written the first time you describe the inoculation method;
Line 150-151: KClO3;
Line 142: this is McKinney index, you should cite the original reference;
Line 190: move the citation of Fig.7 and Fig.S1 in Results, after Fig. 6;
Lines 224-226: the legend of Fig. 2 is that of Fig. 1;
lanes 138-140: specify in which part of corm is observed the discoloration area;
Lanes 231-232: Figure 3B?;
Lane 241: delete which;
Lane 261: paragraph 3.4 and Fig. 5: there is confusion among the isolates showed in the figure and those named in the text;
Lane 263: these are 6 isolates;
Lane 271: which isolate?;
Lanes 281-287: there is a refuse of 3.5 paragraph, the phylogenetic analysis lacks